# A Regional Observational Study on COVID-19-Associated Pulmonary Aspergillosis (CAPA) within Intensive Care Unit: Trying to Break the Mold

**DOI:** 10.3390/jof8121264

**Published:** 2022-11-30

**Authors:** Tommaso Lupia, Giorgia Montrucchio, Alberto Gaviraghi, Gaia Musso, Mattia Puppo, Cesare Bolla, Nour Shbaklo, Barbara Rizzello, Andrea Della Selva, Erika Concialdi, Francesca Rumbolo, Anna Maria Barbui, Luca Brazzi, Francesco Giuseppe De Rosa, Silvia Corcione

**Affiliations:** 1Unit of Infectious Diseases, Cardinal Massaia, 14100 Asti, Italy; 2Department of Surgical Sciences, University of Turin, 10126 Turin, Italy; 3Department of Anaesthesia, Critical Care and Emergency—Città Della Salute e Della Scienza Hospital, Corso Dogliotti 14, 10126 Turin, Italy; 4Department of Medical Sciences, Infectious Diseases, University of Turin, 10126 Turin, Italy; 5Unit of Infectious Diseases, ASO SS. Antonio e Biagio e Cesare Arrigo, 15121 Alessandria, Italy; 6Department of Emergency, Anesthesia and Critical Care Medicine Michele and Pietro Ferrero Hospital, 12060 Verduno, Italy; 7Unit of Laboratory Medicine and Microbiology, Cardinal Massaia Hospital, 14100 Asti, Italy; 8Microbiology and Virology Laboratory, Città Della Salute e Della Scienza Hospital, Corso Dogliotti 14, 10126 Turin, Italy; 9Division of Geographic Medicine, Tufts University School of Medicine, Boston, MA 02111, USA

**Keywords:** COVID-19, Aspergillus, COVID-19-associated pulmonary aspergillosis, fungal infection, pulmonary, aspergillosis, bronchopulmonary, intensive care unit, antifungal agents, bronchoalveolar lavage fluid

## Abstract

The reported incidence of COVID-19-associated pulmonary aspergillosis (CAPA) ranges between 2.4% and 35% in intensive care unit (ICU) patients, and awareness in the medical community is rising. We performed a regional retrospective observational study including patients diagnosed with CAPA defined according to the Modified AspICU Dutch/Belgian Mycosis Study Group and CAPA–EECMM, from five different ICUs, admitted between March, 2020 and September, 2021. Forty-five patients were included. The median age was 64 (IQR 60–72), mostly (73%) males. At ICU admission, the median Charlson comorbidity index was 3 (2–5), and the simplified acute physiology score (SAPS)-II score was 42 (31–56). The main underlying diseases were hypertension (46%), diabetes (36%) and pulmonary diseases (15%). CAPA was diagnosed within a median of 17 days (IQR 10–21.75) after symptoms onset and 9 days (IQR 3–11) after ICU admission. The overall 28-day mortality rate was 58%, and at univariate analysis, it was significantly associated with older age (*p* = 0.009) and SAPS-II score at admission (*p* = 0.032). The use of immunomodulatory agents, *p* = 0.061; broad-spectrum antibiotics, *p* = 0.091; positive culture for Aspergillus on BAL, *p* = 0.065; and hypertension, *p* = 0.083, were near reaching statistical significance. None of them were confirmed in multivariate analysis. In critically ill COVID-19 patients, CAPA acquired clinical relevance in terms of incidence and reported mortality. However, the risk between underdiagnosis—in the absence of specific invasive investigations, and with a consequent possible increase in mortality—and over-diagnosis (case identification with galactomannan on broncho-alveolar fluid alone) might be considered. Realistic incidence rates, based on local, real-life epidemiological data, might be helpful in guiding clinicians.

## 1. Introduction

Invasive pulmonary aspergillosis (IPA) in the setting of severe viral respiratory diseases is a well-known condition, being described in the context of influenza [1], non-influenza respiratory viral infections (i.e., respiratory syncytial virus, and parainfluenza) [2], and now, during the SARS-CoV-2 pandemic [3]. Cases of coronavirus disease 2019 (COVID-19)-associated pulmonary aspergillosis (CAPA) were described from the early phase of the pandemic in the intensive care unit (ICU) patients, and since then, many studies were performed to better outline this subject [4].

Four proposed definitions for CAPA were proposed from the outbreak of the pandemic by White et al. [5], Verweij et al. [6], Koehler et al. [3], and Bassetti et al. [7] (Table 1).

The reported incidence rate of CAPA in ICU patients stood between 0 and 34.3% [4], with a variability highly influenced by the study period, being very low in the first COVID-19 waves and growing progressively over time. Several factors could explain this wide range: different awareness of the clinicians, the increasing use of bronchoscopic diagnostics, different criteria of aspergillosis definition and diagnostic strategies, and host and environmental factors [8,9].

SARS-CoV-2-associated acute respiratory distress syndrome (C-ARDS) [10] might be the main risk factor for CAPA, considering that these patients lack of typical host concurrent factors promoting invasive fungal infections, such as malignancies, neutropenia, history of allogenic stem cells (HSCT), or solid organ transplantation (SOT) [11]. This could be explained by the dysregulation of the immune system and the lung parenchyma tissue damage, favouring bacterial and fungal growth [12]. In addition, the incremental use, according to the progressive new evidence in the context of severe COVID-19, of immunomodulating drugs and corticosteroids also certainly played an important role, not yet fully defined [13,14,15].

What is more concerning about CAPA is the higher mortality rate of these patients compared to others without aspergillus coinfection. Recently, Kariyawasam and colleagues reported in a systematic review that of the 240 patients considered to be diagnosed with CAPA according to available criteria [3,5,6,7], 98 patients survived (40.8%) and 142 patients died (59.2%). Poor outcome predictors were older age, pulmonary and cardiovascular diseases, diabetes, active oncological disease, and solid organ transplantation [4,11,16,17]. However, several studies and literature reviews [4,11,18] failed to correlate any risk factor to mortality. According to the high burden of the diseases, different studies evaluated the impact of antifungal prophylaxis in this setting with the aim of reducing the risk of fungal infections, but so far no robust data are available [19,20].

Recent questions regarding the actual clinical impact of CAPA are emerging in the literature, mainly when defined by the isolated positivity of galactomannan on bronchoalveolar lavage fluid, especially considering the extended therapeutic antifungal need [21]. 

In this multicentre, retrospective analysis, we describe the demographic, clinical, and therapeutic characteristics of patients with CAPA in five ICUs in the Piedmont region in Italy during the period from March 2020 to September 2021. In addition, we evaluate possible predictors of poor outcomes, analysing comorbidities, treatment, and diagnostic strategies and timings.

## 2. Materials and Methods

This was an observational, multicentre, retrospective study to describe the incidence of CAPA in SARS-CoV2 pneumonia patients admitted to the involved ICUs between March, 2020 and September, 2021. The study included five ICUs in Piedmont (Italy), Cardinal Massaia Hospital (Asti; 1 ICU), S.S. Antonio and Biagio Hospital (Alessandria; one ICU), Michele and Pietro Ferrero Hospital (Verduno, Cuneo; one ICU) and Città della Salute e della Scienza Hospital (Turin; two ICUs).

The data sources were the hospital administrative records and the Microbiology Laboratory database. Data acquisition and analysis was performed in accordance with the protocols approved by the local Ethics Committee (Ethics Committee: Comitato Etico Interaziendale A.O.U. Città della Salute e della Scienza di Torino—A.O. Ordine Mauriziano—A.S.L. Città di Torino; Ethics approval number 0031285). Written informed consent was waived according to Italian regulation, due to the retrospective nature of this study. The study was conducted according to the guidelines of the Declaration of Helsinki.

The main objective of this study was to analyse the risk factors associated with the mortality of patients with CAPA in ICUs. All the centres were asked to provide data on demographics, medical history and co-morbidities, risk factors for invasive fungal infections, information about the hospitalisation with details on radiological and microbiological diagnostic, and treatment provided during the stay in ICU and outcomes. Data were extracted and revised from local servers by experienced research physicians in each centre.

We defined CAPA using the Modified AspICU dutch/Belgian Mycosis Study Group [6] and CAPA–EECMM [3] consensus definitions based on clinical, radiological, and microbiological criteria. (Table 2)

All consecutive adult (≥18 years) patients admitted to the ICU for severe SARS-CoV-2 pneumonia and presenting CAPA criteria during the study period were enrolled. All patients were followed-up with until the hospital discharge to compute: ICU, 28-day and overall mortality, as well as length of ICU and hospital stay.

During the study period, bronchoalveolar lavage fluid (BALF) cultures and galattomannan (GM) on BALF or on serum were performed on clinical decision. Surveillance cultures (tracheal aspirate, rectal swab, and urinary culture), performed weekly, did not include galattomannan test or broncho alveolar lavage. Tracheal aspirate comprises both fungal and bacterial cultures.

### 2.1. Microbiology

Detection of the SARS-CoV-2 virus was performed by real-time RT-PCR with Xpert^®^ Xpress SARS-CoV-2 (Cepheid Inc., Sunnyvale, CA, USA).

All the microbiological pulmonary samples were obtained through bronchoalveolar lavage fluid (BALF); no sterile biopsy was conducted.

The galactomannan antigen index was measured with the immunoenzymatic sandwich test (EIA) Platelia Aspergillus Ag (Bio-Rad Lab., Marne-La-Coquette, France) in BALFs and serum specimens. GM cut-offs were set at 1.0 and 0.5 optimal density index (ODI), in BALF and serum, respectively.

Moreover, BALFs were cultured on Sabouraud dextrose agar (SDA, Becton Dickinson GmbH, Heidelberg, Germany) dishes. Fungi were identified by microscopic examination and by matrix-assisted laser desorption ionization time of flight (MALDI-TOF) (MicrofleX, Bruker Co., Billerica, MA, USA).

### 2.2. Statistical Analysis

Data were entered and analysed using SPSS version 27. Descriptive analysis was reported as frequencies and percentages for categorical variables and means and standard deviations for numeric variables.

Dichotomous variables were evaluated against mortality using chi-square test. Statistical significance was defined as less than 0.05. Continuous variables were tested for normality by the Kolmogorov–Smirnov test. Normally distributed variables were evaluated using *t*-tests. Not normally distributed variables were evaluated using Mann–Whitney test.

All variables with *p* ≤ 0.05 in the univariate analysis were considered for inclusion in the multi-variate analysis. Logistic regression was used to identify independent risk factors associated with acquisition of CAPA, with *p* ≤ 0.05 considered to indicate significance.

## 3. Results

We collected a total of 45 patients with a diagnosis of CAPA over the study period. The median age among our patients was 64 years (IQR 60–72); 73% (N = 33/45) of them were male and 29% were active smokers (Table 3).

### 3.1. Baseline Conditions at Admission

At ICU admission, the median Charlson comorbidity index was 3 (IQR, 2–5) and the simplified acute physiology (SAPS)-II [22] score was 42 (IQR, 31–56). The main underlying diseases and baseline characteristics of our population were hypertension (46%), diabetes mellitus (36%), cardiovascular (29%) and pulmonary diseases (15%), and renal failure (18%). Thirty-seven percent were immunodepressed: of those three had haematological diseases, three received solid organ transplantations (7% both), seven had an active malignancy (15%), and five had other causes of immunodepression (Table 4).

Twenty-six percent of our patients were admitted directly in ICU, while 49% were transferred to ICU within seven days from hospitalisation and the last fifteen percent were admitted in ICU after more than seven days of hospitalization. Eighty-nine percent of them (N = 40) underwent mechanical ventilation (MV). During the hospitalization, the majority of our patients (91%, N = 41) received therapy with corticosteroids, while treatment with immunomodulatory agents was administered in 18% (N = 8) of the patients.

IL-6 was tested in 21 patients (46.7%) as a COVID-19 outcome predictor. The mean value was 48.3 (IQR 19.4–163.1).

**Table 3 jof-08-01264-t003:** Descriptive analysis of the population.

Variable	Median	IQR
Age (years)	64	60–72
Charlson score	3	2–5
SPAPS II score	42	31–56

**Table 4 jof-08-01264-t004:** Baseline conditions at the admission.

Variable	N	Rate (%)
Hypertension	21	46
Diabetes mellitus	16	36
Cardiovasculare diseases	13	29
Renal failure	8	18
Pulmunary diseases	7	15
Active malignancy	7	15
Haematological diseases	3	7
Solid organ transplantation	3	7
Other causes of immunodepression	5	11
Liver cirrhosis	1	2

### 3.2. Previous Antimicrobial Therapy

Concerning previous antifungal treatment, nineteen patients (42%) received a pre-emptive antifungal therapy during MV. Among them, 52.5% (10/19) were mould-active drugs: liposomial amphotericin B and isavuconazole were started in six and four patients, respectively; in the other eight patients, echinocandins were administered, probably on the suspect of candidiasis. No statistically significant reduction in the mortality rate was found in those patients who started pre-emptive antifungal therapy.

Considering bacterial super-infections, 36 patients (80%) had documented intercurrent bacterial infections during their ICU-stay and 43 (96%) received broad-spectrum antibiotics.

### 3.3. Diagnostic Procedures

All the microbiological pulmonary samples were obtained through bronchoalveolar lavage fluid (BALF); no sterile biopsy was conducted. Nineteen patients (42%) had cultures positive for *Aspergillus species* on BALF. Positive GM levels were found on BALF in 35 patients (78%) and on serum in 9 patients (20%).

### 3.4. Definition and Diagnosis of CAPA

Considering the CAPA–EECMM [3] and Modified AspICU dutch/Belgian Mycosis Study Group definitions [6], we sorted 43 patients into probable or putative CAPA. Of note, the same patients fulfilled the criteria to be correctly sorted with both the definitions (Table 3). Only two patients failed to be identified as CAPA, even with high suspects for radiological imaging and clinical presentation: in one, GM level on BALF was under the cut-off needed (0.95 ODI) and the other one died with no microbiological isolations.

The median time to CAPA diagnosis from the onset of COVID-19 symptoms was 17 days (IQR 10–21.75), 9 days (IQR 3–11) from the admission in ICU, and 6 days (IQR 0–6.5) from the start of MV.

### 3.5. Antifungal Therapy

Analysing the therapeutic approach to CAPA diagnosis, overall, 32 patients (71%) received an active anti-mould therapy after CAPA diagnosis or suspect. The most common drugs used were azoles, with voriconazole administrated in 34% of the CAPA (N = 11) and isavuconazole in 31% (N = 10). Liposomal amphothericin B was administred in 22% (N = 7) of the patients, and six of them (86%) received it as a first-line treatment. Echinocandins (caspofungin and anidulafingin) were provided to 12.5% (N = 4) of these patients.

Adverse events occurred in only two patients during antifungal therapy, and were represented by renal and hepatic impairment.

### 3.6. Outcome

The overall mortality rate observed in our population, after 7 days from ICU admittance, was 20% (9/45) and reached 58% (26/45) at the 28th day.

We analysed the effects of our population’s clinical characteristics on mortality in patients with CAPA on the 28th day from admission in ICU (Table 5).

At the univariate regression model, older age (*p* = 0009) and higher SAPS-II score at admission (*p* = 0.032) were independently associated with significantly increased overall mortality.

Administration of immunomodulatory agents (*p* = 0.061) or broad-spectrum antibiotics (*p* = 0.091), positive culture for *Aspergillus* on BAL (*p* = 0.065) and hypertension (*p* = 0.083) failed to reach significance as risk factors on the overall mortality (Table 6).

None of the considered factors for overall and 28-day mortality were confirmed in multivariate analysis.

## 4. Discussion

This multicentre, retrospective study evaluated the clinical characteristics and outcomes of critically ill CAPA patients admitted to five different ICUs in the Piedmont region, Italy, adding an essential piece to the growing epidemiological and clinical knowledge of this fascinating condition. Considering possible predictors of poor outcomes, only older age and high SAPS-II scores at admission were independently associated with significantly increased overall mortality in the univariate regression model. However, administration of immunomodulatory agents or broad-spectrum antibiotics, positive culture for Aspergillus on BAL, and hypertension seemed to be correlated with increased mortality at 28 days from admission to the ICU, but failed to reach statistical significance, as with all the considered factors in the multivariate analysis, probably due to the limited sample size. 

In our population, clinical characteristics, such as median age (64) and gender (73% males), are consistent with recent systematic review data published by Kariyawasam et al. (65 years; 67.8%) [4]. Accordingly, the distribution of primary underlying diseases and baseline characteristics in this regional retrospective study overlapped many studies of large numbers of CAPA patients. Cardiovascular diseases and diabetes mellitus, followed by chronic respiratory diseases, were the most frequent comorbidities [8,9,10,11,12,13,14]. In contrast, we found a very high rate of immunosuppressed patients, most of whom had active malignancy (15%). In addition, the proportion of immunocompromised patients considered host factors in some case definitions varied from 0 to 17.6%. However, the denominators calculated by the authors of these publications did not represent the entire COVID-19 ICU patient population, but CAPA subgroups in most cases, according to our study. Moreover, in the Kariyawasam et al. systematic review, approximately 70% of patients received steroid or immunomodulatory therapy, compared to 91% in our population [4].

Critically ill COVID-19 patients are more vulnerable than less-critical patients, in which primary coinfections are rare [16,17]. Developing secondary infections is based on multiple organ failure, prolonged MV time, and dependence on renal replacement therapy or extracorporeal membrane oxygenation [17,18]. Moreover, it is well known that patients with severe SARS-CoV-2 infection admitted to ICUs are more likely to develop CAPA than those with lower oxygen support needs [4,11]. In line with these data, the need for MV was widespread in our population (89%), similarly to the literature (90.7%) [4,10], and it represents a known risk factor for IPA [23] also in non-COVID-19 patients. Additionally, SAPS II score well defined the clinical severity of our population.

Considering the timing of development of CAPA, in our populations, the median time to CAPA diagnosis from admission to ICU was nine days (IQR 3–11) and within the range reported by Chong et al. (e.g., 4–15 days) [24]. Similarly, the time to CAPA diagnosis after MV start (6 days, IQR 0–6.5) was inside the range reported by Chong et al. in their systematic review: between 3 and 8 days [24].

Diagnosis is crucial in invasive pulmonary aspergillosis (IPA), which is in CAPA. Lung biopsy, BALF culture, and GM testing of BALF are the gold standards for IPA diagnosis. However, BALF and autopsy are usually lacking in patients with COVID-19 due to the fear of viral particles spreading [3,4,5,6,7]. Our cohort obtained all microbiological bronchial samples through BALF, even though no sterile biopsy was conducted. Interestingly, 42% of patients with BALF cultures were Aspergillus spp. positive, which is in line with previously reported retrospective analysis (Xu et al. [25], 48.7% positive cultures), while the rate of serum GM in our cohort was lower than that reported previously by Xu et al. (20% vs. 61.5%) [25].

Regarding serological biomarkers, we collected IL-6 in 46.7% of CAPA patients with a median value of 48.3 (IQR 19.4–163.1). Increased levels of IL-6 were also reported in severe cases of COVID-19, impacting immune cell function and the antiviral mechanisms of immune cells [25,26,27]. Considering the role of antifungal therapy in COVID-19 patients with positive Aspergillus tests of their respiratory samples, the literature is inconclusive [4]. The most common drugs used in our population were azoles (voriconazole 34% and isavuconazole 31%), which is in line with ECMM expert guidelines [16]. Furthermore, liposomal amphotericin B was used in 22% of cases, which can be a salvage therapy or even initial therapy if local azole resistance patterns are high and considering BALF penetration, especially in critically ill patients. About 12.5% of CAPA patients received empirical treatment with echinocandins with the suspicion of invasive candidiasis; echinocandins are not recommended as monotherapy, but combined with an azole in areas with a high prevalence of azole resistance. The association between antifungals and mortality in CAPA patients remains unclear despite the widespread use of antifungals in previously reported CAPA studies [4,20,21].

A rising concern is the likelihood that patients with CAPA have invasive aspergillosis and how mortality might be directly related to the fungal infection. Although CAPA was reported in 5–10% of critically ill COVID-19 patients, the incidence varies widely (0–33%) across studies, most cases are unproven, and definitions and clinical relevance are debated [9].

Recently, Clancy et al. questioned the likelihood that patients with CAPA have invasive aspergillosis, using diagnostic test performance in other clinical settings to estimate positive predictive values (PPVs) and negative predictive values (NPVs) of CAPA criteria for invasive aspergillosis in populations with varying CAPA incidence. Authors find that according to the incidence of CAPA in the population, the PPV and NPV have relevant differences and depending on local epidemiology and clinical details of a given case, PPVs and NPVs may be useful in guiding antifungal therapy [9].

Moreover, on the one hand, the diagnostic possibility is closely related to the clinician’s attention and the specific search for fungal infections during bronchoscopy. On the other hand, the impact of the choice of treating all critically ill patients with a positive GM on BALF with antifungal therapy should not be underestimated.

In this sense, the relatively frequent finding of CAPA potentially opens the way to possible strategies for preventive treatments, especially in selected populations [28]. Unfortunately, no statistically significant reduction in the mortality rate was found in those patients who started pre-emptive antifungal therapy. To better understand the usefulness of pre-emptive therapy in CAPA, there is an ongoing trial on using isavuconazole to prevent CAPA (ClinicalTrials.gov Identifier: NCT04707703) that might fill the gap for this issue soon [29]. In our cohort, no patients were found on antifungal prophylaxis, in line with the systematic review by Chong et al. in all included studies [18]. While mold-active antifungal prophylaxis was shown to be successful in preventing CAPA in some single-centre cohort studies, a more considerable level of evidence is needed before this approach can be recommended [19,20].

The overall mortality rate observed in our population after seven days from ICU admittance was 20% (9/45), but it climbed to 58% on day 28. The hospital mortality observed in CAPA patients in the literature was highly variable, ranging between 22.2% and 100% [18].

Considering the impact of clinical characteristics, in the univariate regression model, older age (*p* = 0.009) was independently associated with significantly increased overall mortality. Invasive pulmonary aspergillosis is a fatal disease in the elderly [28], despite COVID-19 coinfection. Apostopoulos et al., in their systematic review, showed that the only factor significantly associated with mortality was age [2], in line with the following meta-analyses [4,18,28,30].

Moreover, in our cohort, a higher SAPS-II score at admission (*p* = 0.032) was independently associated with significantly increased overall mortality, according to previous findings [31], suggesting the relevance of SAPS II as a predictor of mortality, specifically in CAPA patients.

There are several limitations to this study. First, this is a multicentre-centre study of a single Italian region that may not accurately reflect the general demographics of Italy. Moreover, the small sample could not assess any potential evidence of risk factors for mortality for CAPA in general. In addition, the use of steroids and immunomodulatory agents was not standardized over the study period, but changed according to the most recent literature and guidelines.

Even more, our analysis does not consider the denominator of all critically ill patients with COVID-19 admitted to the considered ICUs in the same period, the comparison of which would allow us to determine the real impact of risk factors and CAPA on mortality. Furthermore, since this is a real-life and observational study, the diagnostic investigations were conducted based on the clinical judgment of the treating physicians and not standardized according to a predefined diagnostic protocol. Nevertheless, within the study’s limits, CAPA was diagnosed through two of the four criteria known for CAPA diagnosis, and a proven diagnosis with lung biopsy or autopsy was unavailable. In addition, we did not collect phenotypic and genotypic characteristics of *Aspergillus* spp. from positive culture.

## 5. Conclusions

In critically ill COVID-19 patients, CAPA confirms his clinical relevance in terms of incidence and mortality. However, if the difficulties in definitions were at least in part overcome, the diagnostic and epidemiological ones are still alive. In the critically ill patient in particular, the risk is between underdiagnosis on the one hand, in the absence of specific invasive investigations, and with a consequent possible increase in mortality for patients; and over-diagnosis on the other, defining a case only with a positive GM on BALF [32,33], with a huge impact in terms of antifungal treatment, development of resistance, and need for antifungal stewardship programs.

Realistic incidence rates, based on local, real-life epidemiological data, coming from different realities as those presented in this work, might make the difference in guiding clinicians. Larger studies are needed to better define risk factors, impact on mortality, appropriate decision treatment strategies, and to investigate a potential benefit of antifungal prophylaxis in patients with severe COVID-19.

## Figures and Tables

**Table 1 jof-08-01264-t001:** Definition of CAPA.

Definition	Clinical	Radiological	Mycological
COVID-19-associated pulmonary aspergillosis White [5]	PCR confirmed COVID-19 infection and one of: refractory fever despite at least 3 days antibiotics. Recrudescent fever of at least 48 h despite antibiotics. Dyspnea, hemoptysis, and pleural rub or chest pain. Worsening respiratory function despite antibiotics and ventilatory support.	New infiltrates on chest X-ray or chest CT when compared with admission, including progression of signs attributed to viral infection. Radiological signs typical of invasive pulmonary aspergillosis (nodules, halos, cavities, wedgeshaped and segmental or lobar consolidation) or evidence of sinusitis should be associated with heightened suspicion of fungal disease.	Proven: Histology/microscopy demonstrating dichotomous septate hyphae in tissue; positive culture from tissue. Putative: Nonspecific radiology: two or more positives across different test types, or multiple positives within one test type, from the following: positive culture from NBL/BAL-positive GM-EIA in NBL/BAL (I ≥ 1.0), positive GM-EIA in serum (I ≥ 0.5), positive Aspergillus PCR in NBL BAL or blood; and positive 1-3-β-D glucan in serum/plasma. Radiology typical of IA: one positive mycological test as listed, unless the typical radiological signs can be attributed to a different underlying infection (e.g., lung cancer, alternative infection). In this scenario multiple positive results would be required to attain a diagnosis of putative IPA. Note: given the etiological diversity associated with sinusitis, multiple positive tests from this list are required to attain a diagnosis of putative IPA.
Modified AspICU eDutch/Belgian Mycosis Study Group [6]	One of the following: (A) refractory fever despite 3 days of antibiotic therapy, (B) recrudescent fever of at least 48 h despite antibiotic therapy, (C) pleuritic chest pain/rub, dyspnea, (D) haemoptysis, (E) worsening respiratory failure despite antibiotic therapy and ventilatory support.	Abnormal imaging on chest radiography or chest CT.	Proven: Histopathological/microscopic evidence of septated hyphae with evidence of tissue damage or positive culture from sterile material. Putative (all 3 criteria): (A) positive BAL culture, OR (B) BAL GM >1.0 ODI, OR (C) serum GM >0.5 ODI.
CAPA-European Excellence Centre for Medical Mycology [3]	One of the following: (A) refractory fever despite 3 days of antibiotic therapy, (B) pleuritic chest pain/rub, dyspnea, (C) haemoptysis.	Abnormal imaging on chest radiography or chest CT.	Proven: Histopathological/microscopic evidence of septated hyphae with evidence of tissue damage or positive culture from sterile material. Probable (all 3 criteria): (A) Positive lower respiratory tract specimen on BAL OR (B) BAL GM >1.0 ODI OR (C) serum GM >0.5 ODI OR (D) positive serum and BAL PCR, OR (E) positive serum PCR _ 2 Possible (all 3 criteria): (A) positive non-BAL lower respiratory tract specimen OR (B) positive non-BAL GM >4.5 ODI OR (C) positive non-BAL GM >1.2 ODI_2, OR (D) positive non-BAL GM >1.2 ODI with non-BAL PCR.
EORTC/MSGERC [7]	One of the following host factors: (A) severe neutropenia, (B) allogeneic stem cell/solid organ transplant, (C) corticosteroid therapy (0.3 mg/kg per day for >3 months), (D) haematological malignancy, (E)congenital/inherited/acquired immunodeficiency, (F) treatment with T-cell/B-cell immunosuppressants.	One of the following: (A) dense, wellcircumscribed lesions with/without halo sign, (B) air-crescent sign, (C) cavity, (D) lobar or segmental consolidation	Proven: histopathological/microscopic evidence of septated hyphae with evidence of tissue damage or positive culture from sterile material. Probable (all 3 criteria): (A) Positive direct test (culture/microscopy on sputum, ETA, and BAL or 2 and more positive PCR on either BAL or serum), OR (B) positive indirect test (GM in serum or BAL).

**Table 2 jof-08-01264-t002:** Distribution of CAPA cases accordingly to [3,6] definitions.

Definition	Proven	Probable	Putative	Non-Classifiable
CAPA–EECMM	0	43 (96%)	0	2 (4%)
Modified AspICU dutch/Belgian Mycosis Study Group	0	0	43 (96%)	2 (4%)

Abbreviations: CAPA: COVID-19-associated pulmonary aspergillosis; ICU: intensive care unit; EECMM: European Confederation of Medical Mycology; International Society for Human Animal Mycology.

**Table 5 jof-08-01264-t005:** Patients’ characteristics and risk factors for mortality at 28 days from the admission in ICU of CAPA patients.

Variable	N	Death (%)	Survived (%)	*p*
Respiratory diseases	7	4 (57.1%)	3 (42.9%)	0.970
Active or past smoking	13	9 (69.2%)	4 (30.8%)	0.321
Hypertension	21	15 (71.4%)	6 (28.6%)	0.083
Cardiovascular disease	13	8 (61.5%)	5 (38.5%)	0.745
Diabetes mellitus	16	10 (62.5%)	6 (37.5%)	0.634
CKD	8	5 (62.5%)	3 (37.5%)	0.766
Haematological diseases	3	2 (66.7%)	1 (33.3%)	0.747
SOT	3	3 (100%)	0	0.125
Malignancy	7	5 (71.4%)	2 (28.6%)	0.426
Other ID	8	3 (37.5%)	5 (62.5%)	0.200
Liver cirrhosis	1	0 (0.0%)	1 (100%)	0.387
Steroids	41	24 (58.5%)	17 (41.5%)	0.741
Immunomodulatory agents	8	7 (87.5%)	1 (12.5%)	0.061
Broad-spectrum ATB	43	26 (60.5%)	17 (39.5%)	0.091
Previous antifungal therapy	10	6 (60%)	4 (40%)	0.872
CT scan previous CAPA diagnosis	20	10 (50%)	10 (50%)	0.345
CT at CAPA diagnosis	25	17 (68%)	8 (32%)	0.121
Positive Aspergillus isolation on BAL	19	14 (73.7%)	5 (26.3%)	0.065
Positive GM on BAL	35	21 (60%)	14 (40%)	0.572
Positive serum GM	9	3 (33.3%)	6 (66.7%)	0.097
Voriconazole	11	7 (63.6%)	4 (36.4%)	0.600
Isavuconazole	10	6 (60%)	4 (40%)	0.592
Amphotericin B	7	4 (57.1%)	3 (42.9%)	0.572

Abbreviations: CKD: chronic kidney disease; SOT: solid organ transplantation; CT: computed tomography; CAPA: COVID-19 associated pulmonary aspergillosis; GM: galattomannann; ID: immunedepression; ATB: antibiotic; and BAL: broncho-alveolar lavage.

**Table 6 jof-08-01264-t006:** Patients’ characteristics and risk factors for overall mortality of CAPA patients.

Variable	N	Death (%)	Survived (%)	*p*
Respiratory diseases	7	6 (85.7%)	1 (14.3%)	0.681
Smokers	13	11 (84.2%)	2 (15.4%)	0.622
Hypertension	21	18 (85.7%)	3 (14.3%)	0.370
Cardiovascular disease	13	12 (92.3%)	1 (7.7%)	0.188
Diabetes mellitus	16	14 (87.5%)	2 (12.5%)	0.350
CKD	8	8 (100%)	0 (0%)	0.119
Haematological diseases	3	2 (66.7%)	1 (33.3%)	0.550
SOT	3	3 (100%)	0	0.370
Malignancy	7	6 (85.7%)	1 (14.3%)	0.681
Other ID	8	6 (75.0%)	2 (15.4%)	0.697
Liver cirrhosis	1	0 (0.0%)	1 (100%)	0.613
Steroids	41	33 (80.5%)	8 (19.5%)	0.793
Immunomodulatory agents	8	8(100%)	0 (0%)	0.119
Broad-spectrum ATB	43	35 (81.4%)	8 (18.6%)	0.278
Previous antifungal therapy	10	8 (80%)	2 (20%)	1.00
CT scan previous CAPA diagnosis	20	15 (75.0%)	5 (25.0%)	0.453
CT at CAPA diagnosis	25	22 (88%)	3 (12%)	0.134
Positive Aspergillus Isolation on BAL	19	15 (78.9%)	4 (21.1%)	0.880
Positive GM on BAL	35	28 (80%)	7 (20%)	1.00
Positive serum GM	9	7 (77.8%)	2 (22.2%)	0.852
Voriconazole	11	8 (72.7%)	3 (27.3%)	0.648
Isavuconazole	10	9 (90%)	1 (10%)	0.669
Amphotericin B	7	6 (85.7%)	1 (14.3%)	0.896

Abbreviations: CKD: chronic kidney disease; SOT: solid organ transplantation; CT: computed tomography; CAPA: COVID-19 associated pulmonary aspergillosis; GM: galattomannann; ID: immunedepression; ATB: antibiotic; and BAL: broncho-alveolar lavage.

## Data Availability

Not applicable.

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
