# Peer review of "A Regional Observational Study on COVID-19-Associated Pulmonary Aspergillosis (CAPA) within Intensive Care Unit: Trying to Break the Mold"

_jof, 2022, doi:10.3390/jof8121264_

Round 1
Reviewer 1 Report
line 137 check spelling for dextrose
204 delete "the" before 22%
282 BALF are not pulmonary samples. They are bronchial lumen samples.
343 use a synonime for "Moreover". It was also used in the previous sentence.
Table 4 check spelling for "pulmonary" and "haematology"
Table 6 check numbers. "In other ID" n=8, death=6 and survived=5
"Bronchiectasis in previous CT", empty columns
According to the classifications, the CAPA diagnosis in this work was putative since no tissue biospies or culture were avaliable.
Aspergilosis in ICUs is considered a nosocomial infection but this fact was not mentioned or considered. The need of prevention of this infection should be brought into the text at least in the introduction. Prevention will always be better than treatment.
Reviewer 2 Report
This is a retrospective cohort study on COVID-associated pulmonary aspergillosis from a region of Italy (Piedmont area). The authors clearly describe their diagnostic criteria, statistical methods and limitations of the study. The study adds to the existing literature on CAPA.
The authors comment in the Abstract and Outcome section that association was not confirmed in multivariate analysis. This implies that multivariate analysis was conducted and that the p-value was not significant. Multivariate analysis was not actually performed in this study. Please rephrase.
Abstract
Lines 42-44: “In our population, the mortality was 58%, a result in line with the literature. Moreover, older age and higher SAPS-II scores at admission seemed to be associated with increased mortality in patients with CAPA”. This is re-iteration of the results. Suggest to edit.
Introduction
The authors can consider removing Table 1 which would be more appropriate for a review article.
Methods
SAPS-II score. Please explain what the abbreviation means and provide reference.
Lines 125-127: “Surveillance cultures (tracheal aspirate, rectal swab, urinary culture), performed weekly, did not include galactomannan test or broncho alveolar lavage”.
The authors can clarify: Did patients have weekly bacterial and fungal cultures of tracheal aspirates? Would remove rectal swab and urinary culture as these are irrelevant.
Results
Line 195-196: “GM level was under the cut-off needed (0.95 ODI)”. Was GM done in BALF?
Discussion
Line 264: “COVID-19 critically ill patients represent 20% of all patients”. Do the authors refer to the time that the study was conducted?
Lines 308-310: Please elaborate further on the study by Clancy et al. The meaning of this paragraph is not clear.
Conclusions
Suggest to shorten the first paragraph (lines 357-366). These are general comments and not conclusions drawn by the study.
